# The Vaccine Uptake Continuum: Applying Social Science Theory to Shift Vaccine Hesitancy

**DOI:** 10.3390/vaccines8010076

**Published:** 2020-02-07

**Authors:** Rachael Piltch-Loeb, Ralph DiClemente

**Affiliations:** 1Emergency Preparedness Research, Evaluation, and Practice (EPREP) Program, Division of Policy Translation and Leadership Development, Harvard T.H. Chan School of Public Health, Boston, MA 02115, USA; 2Department of Social and Behavioral Sciences, NYU College of Global Public Health, New York, NY 10003, USA; rjd438@nyu.edu

**Keywords:** vaccine hesitancy, vaccine uptake, measles, theory, risk communication

## Abstract

Vaccines are the optimal public health strategy to prevent disease, but the growing anti-vaccine movement has focused renewed attention on the need to persuade people to increase vaccine uptake. This commentary draws on social and behavioral science theory and proposes a vaccine uptake continuum comprised of five factors: (1) awareness of the health threat; (2) availability of the vaccine; (3) accessibility of the vaccine; (4) affordability of the vaccine; and (5) acceptability of the vaccine to effectively approach this rising challenge.

## 1. Introduction

Vaccines are the optimal public health strategy to prevent disease. However, vaccine effectiveness is contingent on their use. In the United States, there is a growing anti-vaccine movement, and a corresponding reduction in herd immunity, a population-level threshold that limits the likelihood of epidemic transmission in a susceptible group. In the past year (2019), vaccine hesitancy toward the measles vaccine, in particular, corresponded with 1249 reported measles cases, the highest annual number since 1992. Of those cases, 89% were unvaccinated or had an unknown vaccination status, and 86% were associated with outbreaks in under-immunized, tight-knit communities with shared belief systems that do not encourage vaccination [1]. Globally, the World Health Organization’s (WHO) vaccine advisory group has attributed vaccine hesitancy to a complicated set of factors centered on “complacency, inconvenience in accessing vaccines, and lack of confidence” [2].

Vaccine programs can be a victim of their own success: as the number of persons who receives a vaccine increases, perceptions of the risk and impacts of the disease may, as a direct consequence, decrease. Adverse health effects that may arise from the vaccine may also become more familiar to the public than the disease itself. The rising incidence of vaccine preventable diseases in the United States has raised new concerns about strategic ways to effectively combat the anti-vaccination movement, reduce vaccine hesitancy, and consequently, enhance vaccine uptake.

Clinicians and researchers have signaled their willingness to support changes in policy, health communication, and long-term education strategies to reduce the risk of disease for unvaccinated people [3]. Missing from many of these calls to action, and much of the empirical literature on drivers of vaccination, is a focus on understanding the myriad of associated socio-behavioral factors that can affect vaccine uptake. While clearly presenting as a medical problem, vaccine hesitancy is, at its root, a socio-behavioral and cultural phenomenon as evidenced by initial outbreaks in particular population subgroups including the Orthodox Jewish community in parts of New York, Somali-Americans in Minnesota, or the Slavic community and surrounding neighbors in southwest Washington. Shared vaccine hesitancy beliefs may be rooted in religious dogma, parental-choice belief systems, or simply community norms, depending on the geographic location and people [1,4,5].

The application of social and behavioral sciences to public health threats include the study of population characteristics (social class, age, gender, culture, race/ethnicity), individual beliefs, attitudes and behaviors, and cultural and socio-political systems and policies that affect public health threats and their potential solutions. The discipline draws upon psychology, sociology, anthropology, and political science to apply systems theory and models to complex public health challenges. 

In this commentary we discuss how social and behavioral sciences can play an integral role in understanding, predicting, and promoting vaccine uptake. We propose that by building upon the wealth of well-established theories of health and social behavior, we may be able to more precisely identify and effectively target key constructs that adversely influence vaccine uptake, modify these constructs via myriad intervention strategies to reduce hesitancy, and consequently enhance vaccine uptake. We propose that in tandem with the tightening and enforcement of vaccine requirements for school enrolment, already mandated in most states, these strategies can bolster vaccination uptake. Here, we outline and apply a vaccine uptake continuum, which may be useful in conceptualizing vaccine hesitancy and could play a role in helping to confront this complex public health problem. 

## 2. Social Science Theories

The current refusal to accept specific vaccines such as the measles or influenza vaccine is symptomatic of a broader sentiment of vaccine hesitancy. The objective for the public health and medical community is to reduce vaccine hesitancy and enhance vaccination rates. In particular, health behavior change theories are helpful to understand the range of possible factors that could help catalyze health-promoting behavior change, in this case, enhancing vaccine uptake.

There are several well-tested theories available to understand the mechanisms of health behavior including the Health Belief Model (HBM), Transtheoretical Model (TTM) (also known as stages of change), the Theory of Reasoned Action or Planned Behavior, and others [6,7,8]. A common denominator of many of these theories and models is the conceptualization that people engage in an internal decision-making process weighing the pros and cons of taking a specific action, in this case, being vaccinated. Specifically, people cognitively weigh (evaluate) the severity of the health threat they confront and the perceived benefits or harms of taking a specific action related to that health threat. This individual risk assessment, much like a scale, is influenced by many factors including the perceived risk of a disease, the information available about disease transmissibility and severity, the source of the information (is it a trusted source or not), their personal environment, cultural beliefs, and the social context within which they live and interact. In the current zeitgeist, some portion of individuals, unfortunately, have used a decision-making calculus to determine that the personal and social cost of not vaccinating may outweigh the health benefits of vaccination.

## 3. The Vaccine Uptake Continuum

To enhance vaccine uptake requires a clear, coordinated, and concerted public health effort. The objective is to “shift” people’s risk assessment to increase the perceived value of vaccination. To do this, we propose conceptualizing a “continuum for vaccine uptake” that can be used to more precisely understand the personal factors and social forces affecting vaccine hesitancy and guide the development and implementation of more effective vaccine promotion programs. 

The vaccine uptake continuum is comprised of five factors: (1) awareness of the health threat; (2) availability of the vaccine; (3) accessibility of the vaccine; (4) affordability of the vaccine; and (5) acceptability of the vaccine. By systematically addressing each step of the continuum, we can begin to gain a greater understanding of the factors affecting vaccine hesitancy and, more importantly, can begin to conceptualize, develop, implement, and evaluate effective interventions designed to promote vaccine uptake. Below, we review each step of the continuum, drawing on social and behavioral science. 

### 3.1. Enhancing Awareness of the Health Threat 

Awareness of any health threat involves appropriately targeted and tailored public health education and media/social marketing campaigns to disseminate information regarding the probability of risk, severity of disease, and effectiveness of vaccines to reduce the risk of disease. Due to the historic effectiveness of vaccine promotion campaigns, vaccine-preventable diseases such as measles, mumps, rubella, smallpox, diphtheria, pertussis, and polio are uncommon in the United States. The majority of the U.S population is vaccinated, thus, they do not observe a high prevalence of vaccine-preventable diseases in their community. This may lead to a miscalculation of the perception of the risk of a vaccine-preventable disease in comparison to the rare adverse consequences associated with being vaccinated. Messages that discuss the health threat itself need to be disseminated through multiple channels (e.g., TV, interviews, blogs, clinical practice) by trusted information sources including influential community members, religious leaders, and the media. 

### 3.2. Maintaining Availability of the Vaccine

Availability of the vaccine refers to the vaccine being widely disseminated and ready for distribution. Most vaccines are readily available in the United States. Vaccines are often updated to account for antigenic shifts. Physicians’ offices and pharmacies often procure vaccines for distribution to clients, increasing the capacity to access broader segments of the community. The population is informed about vaccine availability traditionally through visits to their medical provider or pharmacy, suggesting that medical providers and pharmacists play a pivotal role in the vaccine distribution and information pathway.

### 3.3. Ensuring Accessibility of the Vaccine

Accessibility requires public health points of dispensing vaccines, access to medical professionals, and other community based points like a local pharmacy, commonly used for flu vaccines and the shingles vaccine. Selection of the most appropriate access points can help increase vaccine uptake. Most childhood vaccines are now administered in accordance with the Centers for Disease Control and Prevention (CDC) schedule by a pediatrician, regularly seen by a parent and child. School nurses often play critical roles in vaccination for both routine and emergency vaccination distribution. These comfortable and familiar environments help facilitate vaccine uptake through continuity and timeliness. However, there is a need to optimize information describing their availability through other, less traditional vaccination sites that provide ease of access for community members. 

### 3.4. Safeguarding Affordability of Vaccine Programs

Affordability of the vaccine means that the vaccine is not cost prohibitive nor is the cost of vaccine administration cost prohibitive. Private insurers cover the costs of vaccination at little or no cost to the individual. Medicaid also covers the cost of vaccines for children. If a person is under insured or uninsured, the Vaccines for Children program covers the cost of all vaccines for children under the age of 19 who qualify for Medicaid, do not have insurance or cannot afford the out-of-pocket insurance costs for vaccines, and/or are Native American or Alaskan Natives. According to the CDC, there are over 44,000 providers who participate in the Vaccines for Children program and state coordinators in every state, meaning that affordability is not a major barrier to vaccination uptake for most children. However, for adults, while a vaccine may be affordable including its administration, it may still be cost prohibitive.

### 3.5. Encouraging Acceptability of the Vaccine

Finally, there is a focus on understanding the acceptability of the vaccine for the population. In this case, acceptability is often not homogeneous, but rather a heterogeneous factor. Vaccine hesitancy is currently a critical issue resulting from a multitude of factors and deeply rooted belief systems [9]. These include religious beliefs, individual liberty, “natural” health approaches, fear of adverse health consequences, and others that justify vaccine hesitancy. The specific type of vaccine hesitancy belief system may be addressed through community communication and cultural sensitivities. Individuals within specific communities—whether related to one’s documentation status, structural racism, economic disparities, or cultural background—have pre-existing belief systems, and therefore specific counter-messages, specifically targeted to address the belief system held by members of this community, may be most effective at enhancing vaccine uptake. Trust-based messaging rather than those that are fear-inducing, from community members and key community stakeholders may be valuable catalysts for promoting the adoption and uptake of vaccines [10,11]. Consideration of why individuals possess anti-vaccine beliefs and designing targeted messages to combat those beliefs, may be instrumental in ultimately shifting individuals’ risk assessment. Without understanding the “why” for individuals, it is difficult to design programs to effectively and efficiently promote vaccine uptake.

## 4. Conclusions

If the future is to look different from the present, public health campaigns need to intensify efforts to effectively work across the vaccine uptake continuum to shift individuals’ and community attitudes, beliefs, and, their decision-making toward vaccine acceptance. This includes enhancing awareness of the health threat prevented by the vaccine, maintaining availability through trusted channels, ensuring accessibility to all populations, safeguarding affordability through national program, and ultimately, encouraging acceptability by countering specific vaccine hesitancy beliefs. By drawing on social and behavioral science theories, the public health and medical community may be better equipped to more effectively shift individual and community risk assessment. 

Author contributions: Conceptualization, R.P.-L. and R.D.; Writing- original draft preparation, R.P.-L.; Writing-review and editing, R.P.-L. and R.D. All authors have read and agreed to the published version of the manuscript.

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
