# Peer review of "The Vaccine Uptake Continuum: Applying Social Science Theory to Shift Vaccine Hesitancy"

_vaccines, 2020, doi:10.3390/vaccines8010076_

Round 1

Reviewer 1 Report

general anti-vaxxer are on an increase. why that .... ? suggestions 1. give epidemiological data on whether an where the anti-vaccination believe is on an increase 2. explain that the rise and success of a vaccination program has three consequences a) the number of diseased cases decreases and b) the awareness of the infection risk decrease but c) the people see only the cases with complications after vaccination. this is a Dilemma of perception 3. give examples for cultural or religious reasons not to vaccinate ,,. indicate the key persons of such subgroups who should be motivated to publicly propagate vaccination 4. give examples for "why" people reject vaccination ... is it unnatural ... is the programme etatistic ... what is better, a campaign or a law. in Germany now measles vaccination is regulated by law.

Author Response

To the editor and reviewers:

Thank you for the thoughtful comments and overall feedback. We believe the comments have strengthened our commentary! We have considered the reviewers’ suggestions and made the corresponding edits. We have also made small typographical edits throughout the manuscript.

Below please find our point by point response to each reviewer. The reviewer’s comments are in plain text while our responses are in italics. Corresponding in-text changes to the manuscript are delineated using italics and then quotation marks. We have submitted a track changes copy of the manuscript along with this document.

____

Reviewer 1

General anti-vaxxer are on an increase. why that .... ?

Thank you to the reviewer for this comment. The reasons for vaccine hesitancy on complex and varied. In response to this comment, we have added the following, citing the WHO.: “Globally, the WHO’s vaccine advisory group has attributed hesitancy to a myriad of factors centered on “complacency, inconvenience in accessing vaccines, and lack of confidence.””

Specific suggestions

Give epidemiological data on whether an where the anti-vaccination believe is on an increase

Thank you for this suggestion. In response, we have highlighted the connection between under vaccination and disease outbreaks by adding the following to the first paragraph: “In the past year (2019), vaccine hesitancy towards the measles vaccine, in particular, corresponded with 1,249 reported measles cases, the highest annual number since 1992. Of those cases, 89% were unvaccinated or had an unknown vaccination status, and 86% were associated with outbreaks in under-immunized , tight-knit communities with shared belief systems that do not encourage vaccination”

Explain that the rise and success of a vaccination program has three consequences a) the number of diseased cases decreases and b) the awareness of the infection risk decrease but c) the people see only the cases with complications after vaccination. this is a Dilemma of perception

Thank you for this suggestion. We have added the following to the second paragraph of the introduction: “Vaccine programs can be a victim of their own success: as the number of persons who receive a vaccine increases, perceptions of the risk and impacts of the disease may, as a direct consequence,   decrease. Adverse health effects that may arise from the vaccine may also become more familiar to the public than the disease itself.”

Give examples for cultural or religious reasons not to vaccinate ,,. indicate the key persons of such subgroups who should be motivated to publicly propagate vaccination

Thank you to the reviewer for calling attention to this. The number of belief systems and subgroups who may not vaccinate is not uniform and complex. We do not aim to capture all of these groups, but provide a few examples based on recent outbreaks. To that end, we have added the following to the third paragraph of the introduction section: “…as evidenced by initial outbreaks in particular population subgroups including the Orthodox Jewish community in parts of New York, Somali-Americans in Minnesota, or the Slavic community and surrounding neighbors in southwest Washington. Shared vaccine hesitancy beliefs may be rooted in religious dogma, parental-choice belief systems, or simply community norms, depending on the geographic location and people”

Give examples for "why" people reject vaccination ... is it unnatural ...

. In response to this, we have added the following, also referenced above: “Shared vaccine hesitancy beliefs may be rooted in religious dogma, parental-choice belief systems, or simply community norms depending on the geographic location and people”

Is the programme etatistic ... what is better, a campaign or a law. in Germany now measles vaccination is regulated by law.

The reviewer raises an interesting point about the role of policymaking in vaccine uptake. It is not clear that there is a “better” solution but rather that multiple approaches both top down and bottom up may be necessary. Tightening of waivers and exemptions in the United States has helped in some states, but these legal avenues are not often enforced and are therefore not enough on their own. We have added the following to the last paragraph of the introduction to address this comment “We propose that in tandem with the tightening and enforcement of vaccine requirements for school enrollment, already mandated in most states, these strategies can bolster vaccination uptake”

Reviewer 2 Report

From the perspective of microbiology and immunology, obligation to make effort for vaccination means individual immunity as well as herd immunity.  In childhood, vaccination can be help retained individual health, and further, it can also support and protect social environment. In introduction, authors should devote to refer “herd immunity”.

Author Response

To the editor and reviewers:

Thank you for the thoughtful comments and overall feedback. We believe the comments have strengthened our commentary! We have considered the reviewers’ suggestions and made the corresponding edits. We have also made small typographical edits throughout the manuscript.

Below please find our point by point response to each reviewer. The reviewer’s comments are in plain text while our responses are in italics. Corresponding in-text changes to the manuscript are delineated using italics and then quotation marks. We have submitted a track changes copy of the manuscript along with this document.

Reviewer 2

From the perspective of microbiology and immunology, obligation to make effort for vaccination means individual immunity as well as herd immunity.  In childhood, vaccination can be help retained individual health, and further, it can also support and protect social environment. In introduction, authors should devote to refer “herd immunity”.

Thank you to the reviewer for this suggestion. We have added the following to the second sentence of the introduction “…and a corresponding reduction in herd immunity, a population-level threshold which limits the likelihood of epidemic transmission in a susceptible group”

Round 2

Reviewer 1 Report

okay